# Detection of Booroola Polymorphism of Bone Morphogenetic Protein Receptor 1b and Embrapa Polymorphism of Growth Differentiation Factor 9 in Sheep in Thailand

**DOI:** 10.3390/ani14050809

**Published:** 2024-03-05

**Authors:** Poothana Sae-Foo, Supawit Triwutanon, Theera Rukkwamsuk

**Affiliations:** Department of Large Animal and Wildlife Clinical Sciences, Faculty of Veterinary Medicine, Kasetsart University, Kamphaeng Saen Campus, Kamphaeng Saen, Nakhon Pathom 73140, Thailand; poothana.sa@ku.th (P.S.-F.); supawit.tr@ku.ac.th (S.T.)

**Keywords:** bone morphogenetic protein receptor-1B (*BMPR1B*), fecundity (*Fec*), growth differentiation factor 9 (*GDF9*), polymerase chain reaction-restriction fragment length polymorphism (PCR-RFLP), sheep

## Abstract

**Simple Summary:**

Both *FecB* and *FecG^E^* mutations were first identified in sheep in Thailand. These two gene mutations have been reported to be associated with multiple births in sheep. To improve crossbred sheep prolificacy, selecting sheep containing the *FecB* allele was one of the better options. In addition, the coexistence of the *FecB* and *FecG^E^* allele mutations was found in sheep, and this requires further study with respect to its effects on ovulation rates and prolificacy.

**Abstract:**

This study aimed to investigate the appearance and frequencies of the Booroola polymorphism of the bone morphogenetic protein receptor 1b (*BMPR1B*) gene (*FecB*) and the Embrapa polymorphism of the growth differentiation factor 9 (*GDF9*) gene (*FecG^E^*) in sheep in Thailand. A total of 454 crossbred sheep blood samples were collected from four provinces in Thailand during August 2022 to July 2023. The polymerase chain reaction-restriction fragment length polymorphism (PCR-RFLP) method was used to identify the *FecB* and *FecG^E^* genotypes. The history of ewe birth types was collected from the owners to analyze the association between fecundity (*Fec*) genotypes and the history of birth types. The genotypic frequencies of *FecB* for homozygous genotype (*B*/*B*), heterozygous genotype (+/*B*), and wildtype (+/+) were 0.22%, 1.54%, and 98.24%, respectively. Meanwhile, the genotypic frequencies of *FecG^E^* for homozygous genotype (*E*/*E*), heterozygous genotype (+/*E*), and wildtype (+/+) were 0.00%, 2.42%, and 97.58%, respectively. Furthermore, three ewes exhibited both *FecB* and *FecG^E^* genotypes. Fisher’s exact test revealed that possession of the *FecB* genotype was associated with multiple births (*p* < 0.01). Both *FecB* and *FecG^E^* mutations were identified in crossbred sheep in Thailand. Sheep containing *FecB* allele could be alternative candidates to be selected to improve the prolificacy of crossbred sheep in Thailand.

## 1. Introduction

The ovulation rate and litter size are genetically affected by many minor genes and some major genes called fecundity (*Fec*) genes [1]. Bone morphogenetic protein receptor 1b (*BMPR1B*) and growth differentiation factor 9 (*GDF9*), located on ovine chromosomes 6 and 5, respectively, are two of the three major *Fec* genes in the transforming growth factor beta (TGF-β) superfamily.

The *FecB* allele of the *BMPR1B* gene is known to be the first major gene associated with prolificity in sheep, found in Booroola Merino breed [2,3,4]. The A-to-G transition at nucleotide position 746 of the cDNA induces a nonconservative substitution of glutamine with arginine at position 249 of the protein (Q249R). This mutation has been hypothesized to be a partial loss-of-function mutation that is less sensitive to the action of bone morphogenetic protein 4 (BMP-4) on the inhibition of progesterone production and the proliferation of granulosa cells [2,5], enabling a higher sensitivity to follicle-stimulating hormone (FSH) sensitivity [6], as well as the earlier acquisition of luteinizing hormone receptors (LH) and LH-induced responses in granulosa cells of antral follicles. Consequences include precocious follicular maturation, i.e., ovarian follicles maturing at much smaller diameters, with each follicle containing fewer granulosa cells than a similar-sized follicle in the wildtype [7]. *FecB* mutation has been reported in many prolific sheep breeds distributed in several countries, including Australia, New Zealand, India, China, Indonesia, and Iran [8]. The weighted mean effects of *FecB* were summarized in ewes with one copy (+/*B*); they increased the ovulation rate by +0.26 ova (+0.8 to +2.0) and the litter size by +0.67 (+0.4 to +1.3) of the born lambs. However, the effect of a second copy (*B*/*B*) compared to wildtype (+/+) ewes was +3.61 for the ovulation rate and +0.77 for litter size [9].

The *GDF9* gene is associated with ovarian folliculogenesis through the organization and proliferation of the ovarian follicle component via an encoding factor that promotes the development of primordial follicles and stimulates the proliferation of granulosa cells. More than 14 positions of the *GDF9* gene mutation in sheep have been reported, but 5 positions are associated with the ovulation rate and/or prolificacy, which are separate in some prolific sheep breeds. These are *FecG^H^* (high fertility) in Belclare and Cambridge [10], *FecG^T^* (Thoka) in Thoka Cheviot sheep [11], *FecG^E^* (Embrapa) in Santa Inês [12], *FecG^V^* (Vecaria) in Ile de France breed [13], and *FecG^NWS^* or *FecG^F^* (Norwegian White Sheep, NWS; Finnish Landrace sheep, F) [14,15]. The *FecG^E^* mutation was first found in the prolific Santa Inês sheep breed in Brazil. The transition from T to G at nucleotide position 1034 of cDNA represented a nonconservative amino acid change at position 345 from phenylalanine to cysteine (F345C). The ovulation rate of homozygous ewes increased by 65% and 82% compared to heterozygous ewes and wildtype ewes, respectively. The prolificacy of homozygous ewes increased by 23.6% and 57.5% compared to heterozygous ewes and wildtype ewes, respectively [12]. Recently, the *FecG^E^* mutation was also found in Mexican Pelibuey sheep [16].

The number of sheep and sheep farms in Thailand has continuously increased, from 42,040 heads and 5170 farms in 2013 to 126,836 heads and 8472 farms in 2022 [17]. Sheep farming in Thailand can be classified into breeder sheep farms, which mainly focus on breeding purebred meat-typed sheep such as Dorper, Santa Inês, and Katahdin, and commercial sheep farms for meat markets, which raise mostly crossbred sheep. To increase the profit and production of sheep farms, increasing reproductive efficiency in sheep farms, such as via litter size (prolificacy), is important. To alleviate the inefficiency and long cycle length of traditional breeding, marker-assisted selection (MAS) for *Fec* alleles through molecular genetics for the genetic improvement of reproduction efficiency was used [18].

According to the distribution of *FecB* reported for several breeds of sheep in several countries, together with the history of raising Santa Inês sheep breed in Thailand, it was questionable as to whether the *FecB* existed in the Thai sheep population. To date, there has been no previous study on fecundity genes in sheep in Thailand; therefore, the present study aimed to determine *FecB* and *FecG^E^* for the first time in the crossbred sheep population in Thailand. The presence of *FecB* and *FecG^E^* genes, which lead to an increase in litter size, could be used as an alternative method to improve the profitability of sheep farming.

## 2. Materials and Methods

### 2.1. Animals and Sample Collection

A total of 454 blood samples of crossbred sheep were drained from 430 female and 24 male sheep raised in 21 sheep farms located in four major sheep-raising provinces in Thailand: Kanchanaburi, Suphanburi, Nakhon Pathom, and Kamphaeng Phet. The sheep’s ages ranged from seven months to four years. Approximately 3 to 6 milliliters of blood samples were drained from the jugular vein of each animal using an aseptic technique then transferred to an EDTA blood collection tube and stored within an ice box during transport to the laboratory. Samples were stored in a −20 °C refrigerator until DNA extraction and PCR-RFLP were performed.

### 2.2. DNA Extraction

DNA extraction from white blood cells was performed with the proteinase-K silica-based membrane column method using the FavorPrep^TM^ tissue genomic DNA extraction mini kit (FAVORGEN Biotech Corporation, Ping-Tung, Taiwan) according to the manufacturer’s instruction protocol.

### 2.3. Primers Designing

The PCR-RFLP primer set for the identification of *FecB* has been cited by Davis et al. [19]. For *FecG^E^*, the set of primers used to differentiate the *FecG^E^* genotype were designed using the website http://primer1.soton.ac.uk/primer1.html (accessed on 14 September 2022); then, the outer set was selected, which can be used for PCR-RFLP (Table 1). The *GDF9 ovine* gene was recovered from the NCBI database (accession number NM_001142888.2). 

### 2.4. Polymerase Chain Reaction (PCR) Condition

Each PCR reaction was performed in 20 µL total volume containing 2 µL of 10X PCR buffer, 200 µM of each dNTP, 0.5 µM of each primer, 50–100 ng of ovine genomic DNA template, and 0.5 U of DreamTaq DNA Polymerase (Thermo Fisher Scientific, Waltham, MA, USA). Optimized annealing conditions for both *FecB* and *FecG^E^* primers were determined via gradient PCR. PCR was performed with the following condition: initial denaturation at 95 °C for 3 min, followed by 35 cycles of denaturation at 95 °C for 30 s, annealing for 30 s at 60 °C for *FecB* or 52 °C for *FecG^E^*, extension at 72 °C for 30 s for *FecB* or 60 s for *FecG^E^*, and final extension at 72 °C for 5 min.

### 2.5. Restriction Fragment Length Polymorphism (RFLP) and Gel Electrophoresis

FastDigest *Ava*II Enzyme (Thermo Fisher Scientific, USA) and FastDigest *TscA*I Enzyme (Thermo Fisher Scientific, USA) were used to differentiate genotypes of *FecB* and *FecG^E^* genotypes, respectively, following the protocol according to the manufacturer’s instructions. Product samples containing *FecB* mutation were digested into 30 and 160 bp fragments, whereas non-carrier products remained uncut at 190 bp. For *FecG^E^*, the *TspR*I enzyme digested the *FecG^E^* PCR product samples of non-carrier sheep into two fragments of 42 bp and 415 bp, while the product samples containing the *FecG^E^* mutation were digested into three fragments of 42, 173, and 242 bp. All digested RFLP products were electrophoresed by running 10 µL of the product through 2% agarose gel stained with RedSafe^TM^ nucleic acid staining solution at 100 volts for 20 min. Fragment-specific sizes were distinguished using the ExcelBand^TM^ 100 bp + 3K DNA ladder (SMOBIO Technology, Inc, Hsinchu, Taiwan). The gel was visualized under ultraviolet (UV) light and imaged to differentiate the genotype of the samples by using the GelDoc Go imaging system (Bio-Rad Laboratories, Inc., Berkeley, CA, USA).

### 2.6. Direct DNA Sequencing of PCR Products

To verify the identified polymorphisms from PCR-RFLP, some *FecB* and *FecG^E^* carrier PCR product samples were submitted for DNA purification and DNA sequencing using the Sanger Method (Celemics, Inc., Seoul, Republic of Korea) by U2Bio (Thailand) Co., Ltd. (Bangkok, Thailand). The nucleotide sequence data and the chromatogram were analyzed by Bioedit v7.2.5 [20].

### 2.7. Data Collection and Statistical Analysis

The history of ewe birth types was collected from the owners to analyze the association between *Fec* genotypes and the history of ewe birth types using Fisher’s exact test. The genotype distribution was analyzed using the Chi-square test to test the deviation from the Hardy–Weinberg equilibrium. All statistical analyzes were analyzed using the R-studio software version 4.1.3 [21].

## 3. Results

### 3.1. PCR-RFLP Results of FecB and FecG^E^

*FecB*-carrying samples displayed a 160 bp band, while noncarrier samples displayed a 190 bp band on gel electrophoresis after RFLP (Figure 1). For *FecG^E^*, carrying samples displayed 42, 173, and 242 bp bands on gel electrophoresis following RFLP, whereas non-carrier samples displayed 42 and 415 bp bands (Figure 2).

The genotypic and allele frequencies of *FecB* and *FecG^E^* from the PCR-RFLP method are shown in Table 2. The genotypic distribution of *FecB* deviated from Hardy–Weinberg equilibrium (*p* < 0.01). On the other hand, the genotypic distribution of *FecG^E^* was in Hardy–Weinberg equilibrium (*p* = 0.97). Furthermore, when combined the *FecB* and *FecG^E^* genotypes, two of these ewes possess both heterozygous *FecB* (+/*B*) and heterozygous *FecG^E^* (+/*E*) genotypes, and one of the ewes possesses both homozygous *FecB* (*B*/*B*) and heterozygous *FecG^E^* (+/*E*) genotypes.

### 3.2. Direct Sequencing of PCR Products

Eight identified *FecB* carrier and three identified *FecG^E^* carrier PCR products were submitted for direct DNA sequencing and the PCR-RFLP results were verified (Figure 3 and Figure 4).

### 3.3. History of Birth Types of Ewe and Association between Fec Genotypes

The history of ewe birth types was derived from the owners, but litter size records of all parity were obtained from only one farm in Kamphaeng Phet Province. The history of Ewe birth types classified by combined *FecB* and *FecG^E^* genotypes is shown in Table 3.

To examine the association between individual *Fec* allele and birth type history, data regarding another *Fec* allele were excluded to separate the influence of *Fec* alleles. Statistical analysis using Fisher’s exact test (Table 4) revealed a statistically significant association between heterozygous *FecB* genotypes (+/*B*) and birth type history (*p* < 0.01). On the contrary, there was no association between heterozygous *FecG^E^* genotypes (+/*E*) and birth-type history (*p* > 0.05).

## 4. Discussion

The PCR-RFLP technique is one of the conventional methods for genotyping single nucleotide polymorphisms (SNPs). The use of the PCR-RFLP technique to genotype *FecB* was first developed by Wilson et al. [3]. The alternative primer set for amplifying the 190 bp PCR product was first used by Devis et al. [19] and subsequently used in several studies. The DNA sequencing results verified the competence of this protocol similar to previous studies. For *FecG^E^*, a new alternative set of PCR-RFLP primers was designed and can be used to differentiate the *FecG^E^* genotype.

The possible origin of the *FecB* allele in crossbred sheep in Thailand could be attributed to two primary causes: the introduction of discharged imported Merino sheep from tourist attractions, and the transportation and trafficking of sheep among Myanmar, China, and Thailand. This later activity could have led to the intermingling of *FecB*-containing ewes from India or China with the local sheep population in Thailand. However, the potential origin of the *FecG^E^* allele in crossbred sheep in Thailand could possibly be related to the introduction of Brazilian Santa Inês sheep since 1997 [22], which were imported into breeder farms. It should be noted that some of these imported sheep possessed *FecG^E^*. Subsequently, these imported sheep were marketed as breeders and bred with the local sheep population in Thailand.

The genotypic distribution of *FecB* in our study resembled several breeds such as Banyabulak [23], Wadi [24], Lori [25], Zandi [8], and Poll Dorset [26]. Our study found that almost *FecB-carrier* ewes exhibited a previous record of multiple births of lambs. This finding corresponds to the summary from Potki et al. [8], that is, that all breeds carrying the *FecB* allele showed a significant relationship between that allele and higher fertility traits, except the Bonapala breed from India. The average effect of *FecB* on litter size could not be determined in our study due to the limited number of *FecB* carrier ewes identified and the limitation of data recording. However, the mean litter size of heterozygous *FecB* (+/*B*) ewe recorded on the farm in Kamphaeng Phet province was 1.33, which was lower than any previous reports. The lower twinning rate of crossbred ewes in Thailand could be due to ewes’ undernutrition, resulting from lower feed quality and poor feed management practices, particularly in free-range commercial sheep farms.

The result that there was no association between *FecG^E^* genotypes and birth type history (*p* = 0.098) in this study corresponds to the result from Silva et al. [12], which found that only the homozygous *FecG^E^* group (*E*/*E*, n = 9) showed a genotype effect on the frequency of twinning per ewe (44%) (*p* = 0.014). The heterozygous *FecG^E^* group (+/*E*, n = 15) did not show differences in the frequency of twinning (14%) compared to wildtype ewes (+/+, n = 15, 0%). Furthermore, they also found that the heterozygous *FecG^E^* group (+/*E*) did not present differences (*p* = 0.612) in the average number of CL (1.34 ± 0.08) or in the frequency of ewes with multiple ovulations (31.8%) compared to wildtype ewes (1.22 ± 0.11 and 14.6%, respectively). However, they found the difference (*p* < 0.001) of the mean prolificacy of the Santa Inês flock not selected for F1 between the heterozygous *FecG^E^* group (*+*/*E*, n = 102) and the wildtype group (+/+, n = 219) which is 1.44 and 1.13, respectively. Therefore, it was possible that the absence of differences in the frequency of twinning between heterozygous and wildtype ewes in this study might be due to the small sample sizes. 

Other possible candidate positions of the *GDF9* gene mutation in sheep in Thailand are *FecG*^1^ or G1 variants, and *FecG^H^* (high fertility) or G8 variants. For *FecG*^1^, G-to-A transition at nucleotide position 260 of Exon 1 caused the amino acid change from arginine to Histidine at residue 87 (R87H). Moradband et al. [27] identified *FecG*^1^ mutation in Baluchi sheep with the genotype frequencies of wildtype, heterozygous, and homozygous being 0.72, 0.20, and 0.08, respectively. However, only heterozygous Baluchi sheep increased litter size (1.386 ± 0.050) compared to wildtype (1.238 ± 0.032) and homozygous carrier ewes (1.032 ± 0.094). In contrast, Liandris et al. [28] discovered that homozygous *FecG*^1^ Chios sheep displayed the highest mean litter size (2.25), and the allelic effect of A allele was 0.33 lambs, but the coexistence of the G and A alleles in the heterozygous ewes result in a statistically significant decrease (negative dominance deviation) in litter size of 0.47 lambs. For *FecG^H^*, the transition from C to T at nucleotide position 1184 of cDNA represented a nonconservative amino acid change at position 395 from serine to phenylalanine (S395F), which replaced an uncharged polar amino acid with a nonpolar one at residue 77 of the mature coding region. The mutations identified in Belclare and Cambridge with frequency carrying of at least one copy were 12.79% and 75.40%, respectively. The estimate effect of *FecG^H^* on increased ovulation rate was 1.79 ± 0.548 (*p* < 0.01) in Belclare ewes and 2.35 ± 0.386 (*p* < 0.001) in Cambridge ewes [10]. Liandris et al. [28] also identified *FecG^H^* in Chios and Karagouniki sheep breeds, but only heterozygous *FecG^H^* Chios ewes had statistically positive dominance deviation (0.44 lambs) based on 26 records of 10 heterozygous ewes. 

From our findings, three ewes possessing *FecB* and *FecG^E^* carriers were discovered on one farm located in Kanchanaburi province. This finding suggests that both *FecB* and *FecG^E^* alleles are intermingled with crossbred sheep in Thailand. The coexistence of *FecB* and *FecG^E^* alleles might potentially increase the ovulation rate together. However, there was no previous report on the influence of coexisting *FecB* and *FecG^E^* alleles on ovulation rate. Further study to compare the ovulation rate between non-carrier ewes and ewes with *FecB*, *FecG^E^*, and the coexistence of *FecB* and *FecG^E^* via estrous synchronization and to perform a laparoscopic examination for counting corpora lutea (CL) to determine the ovulation rate at day eleven after estrous detection may be conducted, along with a prolificacy comparison from the litter size records of these ewes.

## 5. Conclusions

In this study, both *FecB* and *FecG^E^* mutations were identified in crossbred sheep in Thailand. The presence of the *FecB* allele was associated with lambing multiple births, which was consistent with several previous reports. Therefore, sheep that contain the *FecB* allele could be an alternative choice to improve the prolificacy of crossbred sheep in Thailand rather than *FecG^E^*. The discovery of *FecB* and *FecG^E^* carrier sheep suggested that some of the sheep population in Thailand may have crossed over with the Booroola Merino breed or other previously reported *FecB*-containing sheep, as well as the Santa Inês breed for *FecG^E^*. The coexistence of the *FecB* and *FecG^E^* allele mutations was also discovered, and their effect on the ovulation rate and prolificacy should be studied in future research. 

## Figures and Tables

**Figure 1 animals-14-00809-f001:**
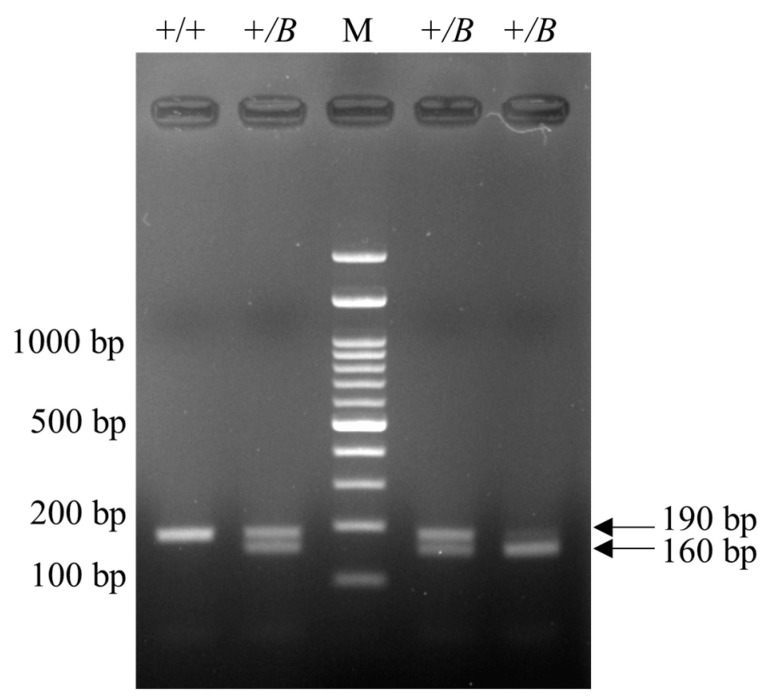
Agarose gel electrophoresis image from PCR-RFLP for *FecB*. PCR-RFLP, polymerase chain reaction-restriction fragment length polymorphism; *FecB*, Booroola polymorphism of bone morphogenetic protein receptor-1b (*BMPR1B*) gene; M, DNA Marker (100 bp + 3K); +/+, non-carrier or wildtype; +/*B*, heterozygous *FecB* genotype.

**Figure 2 animals-14-00809-f002:**
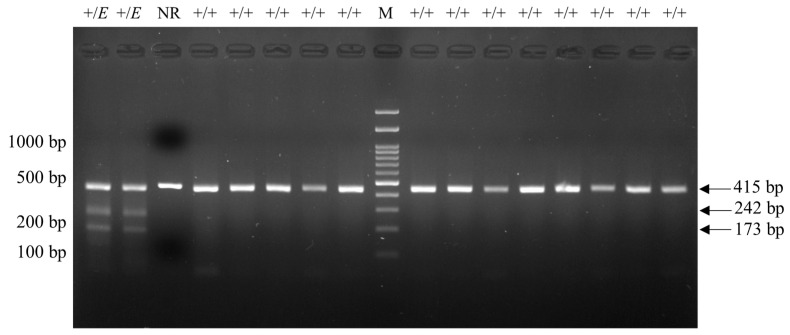
Agarose gel electrophoresis image from PCR-RFLP for *FecG^E^*. PCR-RFLP, polymerase chain reaction-restriction fragment length polymorphism; *FecG^E^*, Embrapa polymorphism of growth differentiation factor 9 (*GDF9*) gene; M, DNA Marker (100 bp + 3K). +/+, non-carrier or wildtype; +/*E*, heterozygous *FecG^E^* genotype; NR, *FecG^E^* PCR product without RFLP.

**Figure 3 animals-14-00809-f003:**
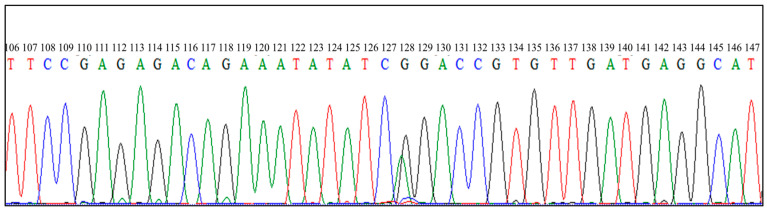
Sequencing result for *FecB* from identified *FecB* carrier. The sample contains both Adenine and Guanine at position 128 representing heterozygous genotype of *FecB* (+/*B*). *FecB*, Booroola polymorphism of bone morphogenetic protein receptor-1b (*BMPR1B*) gene.

**Figure 4 animals-14-00809-f004:**
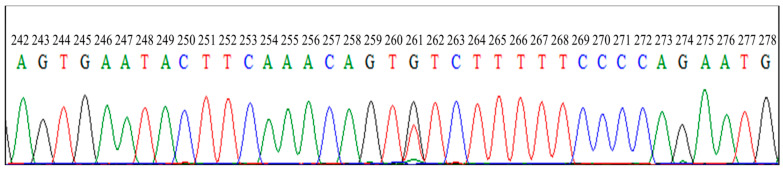
Sequencing result for *FecG^E^* from identified *FecG^E^* carrier. The sample contains Thymine and Guanine at position 261 representing heterozygous genotype of *FecG^E^*. *FecG^E^*, Embrapa polymorphism of growth differentiation factor 9 (*GDF9*) gene.

**Table 1 animals-14-00809-t001:** PCR ^(a)^ primer sets and product size for identification of *FecB* and *FecG^E^*.

Gene	*Fec* Alleles ^(b)^	Primer Name	Primer Sequences (5′-3′)	Product Size
*BMPR1B*	*FecB*	*FecB*-F2 *	CCAGAGGACAATAGCAAAGCAAA	190 bp
		*FecB*-R15 *	CAAGATGTTTTCATGCCTCATCAACACGGTC	
*GDF9*	*FecG* * ^E^ *	*GE*-outF4	CAGCCTGTTTAACATGACTC	457 bp
		*GE*-outR4	GTTCTGCACCATGGTGT	

^(a)^ PCR, polymerase chain reaction; ^(b)^
*Fec*, fecundity; *FecB*, Booroola polymorphism of bone morphogenetic protein receptor-1b (*BMPR1B*) gene; *FecG^E^*, Embrapa polymorphism of growth differentiation factor 9 (*GDF9*) gene. * Taken from Davis et al. [19].

**Table 2 animals-14-00809-t002:** Genotypic and allelic frequencies of *FecB* and *FecG^E^* of crossbred sheep from four provinces of Thailand.

*Fec* Alleles ^(a)^	No. of Animals	Genotypic Frequency ^(b)^ (%)			Allelic Frequency (%)		HWE + ^(c)^
*FecB* (A > G)	454	+/+	+/*B*	*B*/*B*	*B*	+	*p* < 0.01
		446 (98.24)	7 (1.54)	1 (0.22)	9 (0.99)	899 (99.01)	
*FecG^E^* (T > G)	454	+/+	+/*E*	*E*/*E*	*E*	+	*p* = 0.97
		443 (97.58)	11 (2.42)	- (-)	11 (1.21)	897 (98.79)	

^(a)^ *Fec*, fecundity; *FecB*, Booroola polymorphism of bone morphogenetic protein receptor-1b gene; *FecG^E^*, Embrapa polymorphism of growth differentiation factor 9 gene; ^(b)^ +/+, non-carrier or wildtype; +/*B*, heterozygous *FecB*; *B*/*B*, homozygous *FecB*; +/*E*, heterozygous *FecG^E^*; *E*/*E*, homozygous *FecG^E^*; ^(c)^ HWE, Hardy–Weinberg Equilibrium.

**Table 3 animals-14-00809-t003:** Ewe birth types history classified by combined *FecB* and *FecG^E^* genotypes ^(a)^.

Birth Types	Combined *FecB* and *FecG^E^* Haplotypes; n (%)					Total (%)
	+/+, +/+	+/*B*, +/+	+/+, +/*E*	+/*B*, +/*E*	*B/B*, +/*E*	
History of lambing multiple births	9 (60.00)	3 (20.00)	1 (6.67)	2 (13.33)	- (-)	15 (3.49)
Lambing only single birth	380 (98.96)	1 (0.26)	3 (0.78)	- (-)	- (-)	384 (89.30)
Nulliparous	29 (93.55)	1 (3.22)	- (-)	- (-)	1 (3.22)	31 (7.21)

^(a)^ *FecB*, Booroola polymorphism of bone morphogenetic protein receptor-1b *(BMPR1B)* gene; *FecG^E^*, Embrapa polymorphism of growth differentiation factor 9 (*GDF9)* gene; +/+, non-carrier or wildtype; +/*B*, heterozygous *FecB*; *B*/*B*, homozygous *FecB*; +/*E*, heterozygous *FecG^E^*; *E*/*E*, homozygous *FecG^E^*.

**Table 4 animals-14-00809-t004:** Fisher’s exact test results between *Fec* genotypes and birth type history compared with non-carrier.

*Fec* Allele ^(a)^	Genotype ^(b)^	History of Lambing Multiple Births	Lambing Only Single Birth	*p*-Value	Odds Ratio	95% CI ^(c)^
Non-carrier	+/+	9	380			
*FecB*	+/*B*	3	1	<0.01	117.5	8.6, 6295.3
*FecG^E^*	+/*E*	1	3	0.098	13.7	0.2, 192.6

^(a)^ *Fec*, fecundity; *FecB*, Booroola polymorphism of bone morphogenetic protein receptor-1b (*BMPR1B*) gene; *FecG^E^*, Embrapa polymorphism of growth differentiation factor 9 (*GDF9*) gene; ^(b)^ +/+, non-carrier or wildtype; +/*B*, heterozygous *FecB*; +/*E*, heterozygous *FecG^E^*; ^(c)^ CI, confidence interval.

## Data Availability

The data presented in this study are available on request from the corresponding author. The data are not publicly available because the farmers may choose to retain certain data proprietary to maintain a competitive advantage [insert reason here].

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
