# Peer review of "Detection of Booroola Polymorphism of Bone Morphogenetic Protein Receptor 1b and Embrapa Polymorphism of Growth Differentiation Factor 9 in Sheep in Thailand"

_animals, 2024, doi:10.3390/ani14050809_

Round 1

Reviewer 1 Report

Comments and Suggestions for Authors

The main question addressed by the research is the analysis of the association between Fec genotypes (specifically FecB and FecGE) and the history of ewe birth types in crossbred sheep in Thailand. The study aims to investigate the genotypic distribution of key genetic mutations and their potential impact on sheep prolificacy and ovulation rates. The investigation of the presence and frequencies of FecB and FecGE mutations in sheep in Thailand is both original and relevant in the field of animal genetics and reproduction. This research addresses a specific gap in the field by focusing on the genetic factors influencing sheep prolificacy and ovulation rates in a specific geographical region, Thailand. The identification of these mutations and their association with multiple births provides valuable insights for sheep breeding programs and genetic selection strategies aimed at enhancing flock productivity. This research contributes to the subject area by offering new insights into the presence and frequencies of FecB and FecGE mutations in crossbred sheep in Thailand. The study's findings contribute to the existing literature by highlighting the association between these genetic mutations and sheep prolificacy in a specific regional context. Additionally, the identification of FecB and FecGE carriers in the sheep population of Thailand suggests potential opportunities for genetic improvement strategies to enhance reproductive efficiency in sheep farming. The conclusions drawn in the research study are consistent with the evidence and arguments presented throughout the paper. The study successfully addresses the main question posed regarding the association between Fec genotypes and the history of ewe birth types in crossbred sheep in Thailand. The findings support the notion that the presence of the FecB allele is associated with multiple births of lambs, emphasizing its potential role in improving the prolificacy of crossbred sheep in Thailand. The identification of both FecB and FecGE carriers further suggests opportunities for enhancing ovulation rates in sheep populations. The references provided in the research paper are appropriate and relevant to the topic of genetic mutations influencing sheep prolificacy and ovulation rates. The references cited include key studies on Fec genotypes, mutations in bone morphogenetic protein receptor genes, and their impact on ovulation rates in various sheep breeds. These references support the research findings and contribute to the scientific background of the study. For these reasons, it is appropriate to publish the article in its current form.

Reviewer 2 Report

Comments and Suggestions for Authors

Sae-Foo et al. present promising research work on the existence and frequency of genetic polymorphisms, namely FecB and FecGE mutations, in sheep in Thailand. These are commonly associated with the birth of multiple lambs per pregnancy. In this way, the authors intend to contribute to increasing knowledge about sheep selection with the aim of improving the prolificacy of these animals in Thailand.

 In our opinion, studies such as the one presented by Sae-Foo et al. are of great scientific importance. But they are also of great practical importance with a view to improve the profitability of herds, where prolificacy is a very necessary factor. We believe that studies like this are important contributions to the economic development of the populations that raise these animals, which is subsequently reflected in better social development.

 However, in our opinion, we consider that the low frequency of mutations found in the study makes it difficult to associate them with the prolificacy of Thai sheep.

We think that this study may be of more interest from the point of view of describing the appearance and frequency of these mutations in the Thai sheep population. And the exploration they carried out into the possibility of these mutations entering these sheep seems to us to be the most interesting route for this study.

 One aspect that the authors should improve is to increase the number of bibliographic references, so that learning from other studies can help to mature this work.

Comments on the Quality of English Language

In our opinion, minor English revision would be appreciated.

Line 38 – delete “the”, “Bone morphogenetic…”

Line 50 – delete “the”, Consequences…

Line 53 – delete “the”, FecB mutation…

Line 67 – “Inés” correct to “Inês”

Line 91 – blood samples were drained.

Line 94 – were drained from

Line 95 – stored within an ice box

Line 96 – Delete “the”, Samples were stored…

Line 99 – delete “from blood sample”, “from white blood cells”…

Line 124 – “follow the protocol” – “followed…”

Line 125 – will be? Maybe write in the past “products mutation were digested”

Line 128 – same comment to line 125, past sentence

Table 1 – (a) should be next to PCR

Table 2 – legend… BMPR1B where it is described in table?

Table 2 – No. of animals (animals try to write it in the same line)

Title 2.5, paragraph so it can be next to the text related to

Reviewer 3 Report

Comments and Suggestions for Authors

This manuscript delves into the influence of fecundity genes (Fec) on the reproductive efficiency of sheep, focusing on the BMPR1B (FecB) and GDF9 (FecGE) genes. It underscores the significance of these genes in regulating ovulation rate and litter size, with an emphasis on mutations associated with these genes. Additionally, the study investigates the distribution of these mutations in sheep of different breeds and countries, including their presence in sheep populations in Thailand. By employing genotyping techniques and statistical analysis, researchers identified the presence of FecB and FecGE alleles in crossbred sheep in Thailand, suggesting potential for enhancing the prolificacy of these flocks. The findings indicate that sheep carrying the FecB allele may represent a more promising choice for increasing prolificacy compared to FecGE. Moreover, the study highlights the need for further research on the effects of coexisting FecB and FecGE alleles on ovulation rate and prolificacy.

Overall, the manuscript provides a comprehensive overview of the genetic factors influencing ovulation rate and litter size in sheep, with a specific focus on the FecB and FecGE genes. The authors effectively outline the molecular mechanisms underlying the actions of these genes and provide evidence supporting their roles in regulating reproductive traits.

The manuscript presents a study that is well-described and conducted. I believe it could be accepted for publication as an article; however, I have some considerations regarding the current manuscript that could greatly improve it.

Q1. Some statements, such as the association between the presence of FecB allele and multiple births of lambs, need further clarification. While the manuscript suggests that sheep containing the FecB allele are a better choice for improving prolificacy in crossbred sheep, the underlying evidence and potential implications of this assertion could be more explicitly discussed.

Q2. The manuscript proposes intriguing hypotheses about the coexistence of FecB and FecGE alleles and their potential synergistic effects on ovulation rate and prolificacy. However, the discussion of future research directions could be expanded to include specific experimental designs and methodologies for investigating these hypotheses.

In summary, while the manuscript provides valuable insights into the genetic determinants of sheep reproduction, addressing these areas of improvement will enhance the clarity and robustness of the findings, thereby contributing to the advancement of scientific knowledge in this field.

Reviewer 4 Report

Comments and Suggestions for Authors

The manuscript needs revision. Please refer to comments given in the text of reviewed attached file of the manuscript.

Round 2

Reviewer 2 Report

Comments and Suggestions for Authors

The authors made corrections and provided explanations that we consider important. The current version has been improved in our opinion and can be accepted for publication.

Reviewer 4 Report

Comments and Suggestions for Authors

The manuscript can be accepted for publication